# OpenReview forum: "GenARM: Reward Guided Generation with Autoregressive Reward Model for Test-Time Alignment"
_ICLR.cc/2025/Conference — ICLR 2025 Poster_

### Official Review · Reviewer_S14E · 2024-10-23

**Soundness:** 3
**Presentation:** 4
**Contribution:** 3
**Rating:** 8
**Confidence:** 4

**Summary:**

This paper presents GenARM, a novel reward model parameterization for test-time alignment. Comparing with prior works in the domain of test-time alignment, this paper proposes to use the sum of auto-regressive log probability of each token to be the reward model for the entire trajectory. The training objective function and usage for constrained decoding remains the same. Theoretical analysis has been provided to justify this parameterization of the reward model. Empirically, strong performances have been observed for GenARM against other popular baselines such as ARGS and Transfer-Q over three different settings including human preference alignment, weak-to-strong guidance, and multi-objective alignment.

**Strengths:**

The clarity of the writing is worth mentioning, I enjoyed following the reading. Each component is properly motivated. And the insights from the experiments are properly organized.

The approach is, as far as I am aware of, novel. It also seems easy to train and does not incur too much additional computational overhead.

Theoretical analysis has been provided for completeness of the paper.

The effectiveness of the proposed method is validated thoroughly in the experiments section over three different settings including human preference alignment, weak-to-strong guidance, and multi-objective alignment. The section of weak-to-strong guidance is particularly informative.

**Weaknesses:**

The training objective of the reward model (Equation [5]) draws an interesting connection with DPO objective (Equation [7] in DPO). This might result in an unfair comparison with other test-time alighment methods such as ARGS because now that the training cost of the reward model is basically the same as training-time alignment methods like DPO.

In order to provide a more fair comparison, it might be helpful to dedicate additional experiments/paragraphs illustrating the number of samples/training time comparison with other baselines such as ARGS and Transfer-Q

There is an inconsistency of baselines across different settings. In particular, why is BoN of included in Section 6.1 and Transfer-Q not included in Section 6.2?

**Questions:**

See weakness, it would be great if the authors can explain the connection of the proposed method with DPO objective

---

> ### Author Response · Authors · 2024-11-21
>
> We thank Reviewer S14E for the detailed feedback. We are encouraged that the reviewer finds our writing clear, our approach novel and our experiments comprehensive and informative. We address concerns below.
>
> ---
>
> >Weakness 1:.. This might result in an unfair comparison with other test-time alighment methods such as ARGS because now that the training cost of the reward model is basically the same as training-time alignment methods like DPO.
>
> To clarify, the reward model training costs of our method and other test-time alignment baselines, such as ARGS, are almost **the same**, ensuring a fair comparison. In all of our experiments, both the Autoregressive RM (used by GenARM) and trajectory-level RM (used by baselines like ARGS) are initialized from the same 7B LLM. Importantly, other test-time alignment methods like ARGS also require training a 7B trajectory-level RM.
>
> For a more detailed analysis, including training time comparisons, please refer to our response to Weakness 2.
>
>
> > the training cost of the reward model is basically the same as training-time alignment methods like DPO.
>
> This is not accurate: in our weak-to-strong guidance experiments (Section 6.2), where test-time alignment methods use small RM to guide larger LLM, the training costs of the 7B Autoregressive RM and 7B trajectory-level RM are significantly lower than those of training-time baselines, which require training much larger LLMs, such as a 70B model.
>
> >Weakness 2:In order to provide a more fair comparison, it might be helpful to dedicate additional experiments/paragraphs illustrating the number of samples/training time comparison with other baselines such as ARGS and Transfer-Q.
>
> Thank you for the suggestion! We have updated our paper to include a detailed training time comparison in Appendix C.1 (line 803–817). To summarize, training Autoregressive RM only introduces **negligible increase** in training time, compared with trajectory-level RM: We compared them under identical conditions. Both models were trained on the filtered HH-RLHF dataset using LoRA fine-tuning for one epoch, and both were initialized from the same LLaMA-7B model.
>
> The results are:
> * Training trajectory-level RM took 11,896 seconds
> * Training Autoregressive RM took 12,162 seconds (only 2% more than above)
>
> This demonstrates that training Autoregressive RM introduces a minimal increase in time compared to a trajectory-level RM of the same size. This is expected, as both models are 7B, with the only difference being in the last layer, which adds just 1% more parameters (trajectory-level RM: 6.61B vs. Autoregressive RM: 6.72B).
>
> >Weakness 3: There is an inconsistency of baselines across different settings. In particular, why is BoN of included in Section 6.1 and Transfer-Q not included in Section 6.2?
>
> Thank you for pointing this out! In Section 6.1, we have updated our paper to include results for BoN (N=16). The results show that GenARM achieves a 57.22% win rate and a 60.55% win + 0.5 tie rate against BoN. This demonstrates that GenARM outperforms BoN while being significantly more efficient, as BoN requires generating N samples from the base LLM.
>
> We did not include Transfer-Q in Section 6.2 (Weak-to-Strong Guidance) due to its prohibitive inference time. Transfer-Q generates the full response before predicting the next token, making it extremely slow for scenarios with longer response lengths and larger models. On the AlpacaEval2 benchmark, responses are lengthy (e.g., over 1000 tokens), and in our weak-to-strong guidance experiments, we use larger base LLMs, such as 70B models. Based on our estimates, running Transfer-Q in this setting would take over 2000 hours, making it infeasible for evaluation.
>
> >Question: it would be great if the authors can explain the connection of the proposed method with DPO objective
>
> Thank you for the question! When training an LLM with DPO, the reference policy (i.e., the base LLM) must be pre-specified during training. This means the alignment is tightly coupled to the specific base LLM used during the training phase.
>
> In contrast, when training the Autoregressive RM in our method, no base LLM is used during training. Instead, the trained Autoregressive RM can be flexibly combined with different base LLMs during test-time, offering significant configurability. For example:
>
> * Pairing a smaller Autoregressive RM with a larger base LLM for weak-to-strong guidance.
> * Using multiple Autoregressive RMs to guide a single base LLM for multi-objective alignment.
>
> The key distinction lies in test-time flexibility: DPO ties the alignment process to a specific base LLM during training, whereas our method decouples the RM training from the base LLM, enabling diverse test-time applications and configurations.
>
> ---
> Thank you again for your time and effort in reviewing our paper! Please let us know if the above explanations do not address your concerns. We are happy to answer any further questions.

---

> > ### Comment · Reviewer_S14E · 2024-11-23
> >
> > Thanks for your rebuttal. It has addressed most of my concerns, and I have raised my score.

---

> > > ### Author Response · Authors · 2024-11-24
> > >
> > > Thank you for your response! We are glad that your concerns have been addressed.

---

### Official Review · Reviewer_DM7h · 2024-10-26

**Soundness:** 3
**Presentation:** 3
**Contribution:** 3
**Rating:** 6
**Confidence:** 4

**Summary:**

Test time alignment, which guide frozen LLMs during test time text generation, can align LLMs with preference without incurring training cost.
To address the mismatch between trajectory-level RMs and autoregressive text generation, this paper proposes GenARM, a test-time alignment approach that leverages the Autoregressive Reward Model to predict next-token rewards for efficient and effective autoregressive generation.
Experimental results and theoretical arguments are provided to justify the proposed approach, especially compared to the prior test-time alignment baselines.

**Strengths:**

1. The idea of addressing the mismatch between trajectory-level RMs and autoregressive text generation is promising, and have been gaining attention in the community.
2. The proposed method is relatively versatile in experiments, allowing both standard generation, weak-to-strong guidance, and multi-objective alignment.

**Weaknesses:**

1. Autoregressive Reward Model, trained by aligning the accumulated token-level rewards over a full response with the ground truth preference ordering, has been clearly proposed in the literature, e.g., [1,2] and the reference therein. Further, the idea of per-step-reward guided generation has been presented in [3] and the reference therein. The authors ought to have a adequate citation, discussion, and ideally comparison with these prior works. Otherwise, the contribution of this work will be significantly impaired. In particular, I strongly suggest the authors to move part of the current Related Work to the appendix for discussing in details these prior efforts for dense rewards in LLMs' RLHF/generation.

[1] Feng, Yihao, et al. "Fantastic Rewards and How to Tame Them: A Case Study on Reward Learning for Task-oriented Dialogue Systems." The Eleventh International Conference on Learning Representations.

[2] Yang, Shentao, et al. "Preference-grounded token-level guidance for language model fine-tuning." Advances in Neural Information Processing Systems 36 (2023).

[3] Li, Bolian, et al. "Cascade reward sampling for efficient decoding-time alignment." arXiv preprint arXiv:2406.16306 (2024).

2. The calculation of token sampling probability eq. (7) will roughly double the decoding time, especially compared to train-time alignment methods.

3. Source code seems not anonymously released for review purpose.

**Questions:**

1. How would the per-step reward parametrization, eq. (4), mitigate the length bias that longer responses have more term to be sum up, and hence may automatically have a higher aggregate sequence-level reward?
2. In Figure 3 right, why does the word "words" also receives a low reward?

---

> ### Author Response · Authors · 2024-11-21
>
> We thank Reviewer Dm7h for the detailed feedback. We are encouraged that the reviewer finds our work on addressing the mismatch between trajectory-level reward models (RM) and autoregressive text generation both promising and timely. We also appreciate the reviewer's recognition of our method's versatility, enabling a wide range of applications, including standard text generation, weak-to-strong guidance, and multi-objective alignment. Below we address the concerns in details.
>
> ---
>
> >Weakness 1.1: Autoregressive Reward Model, trained by aligning the accumulated token-level rewards over a full response with the ground truth preference ordering, has been clearly proposed in the literature, e.g., [1,2] and the reference therein.
>
> Thank you for highlighting these relevant works on dense rewards! We have updated our related work section (Section 2, line 114-137) to incorporate and discuss these contributions, as detailed below:
>
> We acknowledge that [1,2], highlighted by the reviewers, are pivotal works leveraging dense, token-level reward signals for **training** LLM policies with reinforcement learning (RL). These approaches tackle the well-known challenges of learning from sparse and delayed reward signals in RL [4]. However, our work differs in several key aspects:
>
> 1. Our paper focuses on **test-time** alignment, specifically without requiring any training of the LLM policy network. Unlike [1,2], which use dense reward signals for RL training, we leverage a token-level RM to enable efficient next-token prediction directly at test time.
> 2. Theoretically, we have proven that the parametrization of our proposed Autoregressive RM can guide frozen LLMs toward any distribution achievable by traditional RMs within the KL-regularized RL framework. This result is **fundamental for enabling test-time alignment** and is a capability **not addressed** in prior works that focus on providing dense rewards for training.
>
> >Weakness 1.2: Further, the idea of per-step-reward guided generation has been presented in [3] and the reference therein.
>
> We have revised our paper to include CARDS [3] in the related work section (Section 2, line 114-137) and compare it with our method in the experiments (Section 6.1: Table 1 and Table 2). Unlike our approach, [3] **does not use a token-level RM** to guide text generation. Instead, it repeatedly generates small semantic segments (a few tokens) and evaluates them using a **trajectory-level RM** to select the most satisfactory option. Similar to ARGS [5] (which we already cited and compared with in the original version), [3] relies on trajectory-level RMs to evaluate partial responses, which can be inaccurate. Moreover, it induces significant inference costs as it needs to generate several candidate segments before choosing one. In contrast, our method is more efficient during inference, as the Autoregressive RM directly predicts the next-token reward for next-token generation.
>
> Experimentally, GPT-4 evaluation on the HH-RLHF dataset demonstrates that our proposed GenARM achieves a **higher win rate** compared to CARDS [3], as summarized below:
>
> | Method | Win | Tie | Lose | Win + $\frac{1}{2}$Tie |
> |-|-|-|-|-|
> |GenARM VS CARDS|54.67|5.22|40.11|57.27|
>
> Moreover, GenARM is significantly **more efficient** than CARDS during inference, as evidenced by the inference time required to generate 128 tokens using a 7B base LLM and a 7B RM, as detailed below.
>
> |  | ARGS | GenARM | Transfer-Q | CARDS |
> |-|-|-|-|-|
> |Inference time (s) |7.74|7.28|130.53|87.09|
>
> Therefore, our conclusion remains unchanged: our method achieves the highest win rate and the lowest inference cost among all test-time alignment approaches.
>
> >Weakness 1.3: The authors ought to have a adequate citation, discussion, and ideally comparison with these prior works. In particular, I strongly suggest the authors to move part of the current Related Work to the appendix for discussing in details these prior efforts for dense rewards in LLMs' RLHF/generation.
>
> Thank you for the suggestions! We have added these works on dense rewards to our related work section (Section 2, line 114-137) and included [3] in our experimental evaluation (Section 6.1: Table 1 and Table 2). To accommodate these additions, as suggested, we have moved part of the related work to Appendix A.
>
>
> References:
> [1] Feng, Yihao, et al. "Fantastic Rewards and How to Tame Them: A Case Study on Reward Learning for Task-oriented Dialogue Systems." ICLR 2023
> [2] Yang, Shentao, et al. "Preference-grounded token-level guidance for language model fine-tuning." Neurips 2023.
> [3] Li, Bolian, et al. "Cascade reward sampling for efficient decoding-time alignment." arXiv preprint arXiv:2406.16306 (2024).
> [4.] Ng, Andrew Y., Daishi Harada, and Stuart Russell. "Policy invariance under reward transformations: Theory and application to reward shaping." Icml. Vol. 99. 1999.
> [5] Khanov, Maxim, Jirayu Burapacheep, and Yixuan Li. "ARGS: Alignment as Reward-Guided Search." ICLR (2024).

---

> ### Author Response · Authors · 2024-11-21
>
> >Weakness 2: The calculation of token sampling probability eq. (7) will roughly double the decoding time, especially compared to train-time alignment methods.
>
> Test-time alignment methods, as demonstrated in our paper, offer unique advantages such as convenient multi-objective alignment and weak-to-strong guidance, which training-time alignment methods struggle to achieve. Therefore, when evaluating inference time, the comparison should primarily be made against other test-time alignment methods. Importantly, our method achieves the **lowest inference time among all test-time methods**, as shown in Table 2.
>
> >will roughly double the decoding time
>
> This is not accurate: when the reward model (RM) is significantly smaller than the LLM (e.g., RM is 7B and LLM is 70B in our weak-to-strong guidance experiment), our method introduces only marginal inference overhead (e.g, 10%) compared to training-time methods. Furthermore, for very large LLMs (e.g., 70B), where training becomes prohibitively expensive, training-time methods are no longer viable. In such cases, our method remains an efficient solution since it only requires training the smaller RM, making it both practical and cost-effective.
>
> ---
>
> >Weakness 3: Source code seems not anonymously released for review purpose.
>
> We have uploaded the code for review purpose.
>
> ---
> >Question 1: How would the per-step reward parametrization, eq. (4), mitigate the length bias that longer responses have more term to be sum up, and hence may automatically have a higher aggregate sequence-level reward?
>
> It is not true that longer responses automatically yield a higher or lower aggregate sequence-level reward. To illustrate, consider the following two responses:
>
> * I love apples [EOS] (shorter response)
> * I love apples for dinner [EOS] (longer response)
>
> Denote token-level reward as r (in our case it is a log probability.) There is no guarantee that r(EOS|I love apples) is higher or lower than r(for|I love apples) + r(dinner|I love apples for) + r(EOS|I love apples for dinner). Furthermore, our training objective in eq. (5) forces the model to assign higher score on prefered response, regardless of the length.
>
>
> This principle also holds in **traditional text generation**: While the log probability of a text sequence is computed as the sum of individual token log probabilities, next-token sampling does not inherently favor longer or shorter responses.
>
> ---
> >Question 2: In Figure 3 right, why does the word "words" also receives a low reward?
>
> Good question!
>
> This RM trained for harmlessness is designed to distinguish harmful tokens from harmless ones. The reward values are meaningful only when comparing the next tokens conditioned on the same prefix; **absolute reward values are not directly interpretable**.
>
> To clarify, we conducted a simple quantitative test with the following examples:
>
> * An effecitve way ... use cruel *words* (original sentence in Fig. 3)
> * An effecitve way ... use cruel *defamation* (new sentence)
>
> While "words" has a seemingly low absolute reward of -1.94, the more harmful token "defamation" receives an even lower reward of -3.21. This demonstrates that during guided generation, the RM correctly assigns a lower probability to generating "defamation," which aligns with its goal of discouraging harmful language. This behavior is exactly what we expect from a token-level RM focused on harmlessness.
>
>
> ---
> Thank you again for your time and effort in reviewing our paper! Please let us know if the above explanations do not address your concerns. We are happy to answer any further questions.

---

> > ### Author Response · Authors · 2024-11-24
> >
> > We sincerely thank the reviewer for your time and effort in reviewing our paper! We have addressed the specific points you raised regarding related work on dense rewards, inference costs of our method and the code, along with questions about our work.
> >
> > Please let us know whether we have addressed your concerns. We are more than happy to provide additional clarifications if you have further questions. Thank you!

---

> > > ### Comment · Reviewer_DM7h · 2024-11-24
> > > **Response to authors**
> > >
> > > Dear authors,
> > >
> > > Thank you so much for your detailed responses and efforts in revising the paper and uploading the code. I have adjusted my rating accordingly.

---

### Official Review · Reviewer_Jvd2 · 2024-10-30

**Soundness:** 4
**Presentation:** 4
**Contribution:** 3
**Rating:** 6
**Confidence:** 4

**Summary:**

The paper looks at reward models for RLHF and suggests training a token-level reward model from human preferences by parametrizing it as a sum of log probabilities. They prove that although it constrains the reward models to belong to a specific model class, this class is rich enough to model any reward under the Bradley-Terry preference framework. Empirically, the authors use the token-level reward to perform inference-time alignment.

**Strengths:**

- The authors identify an important pain point of existing work, such as ARGS, that performs inference-time alignment on a token level without being trained for it.
- The theoretical analysis provides an important insight into the capacity of the parameterization the authors chose.
- The paper is written in a clear and easy-to-follow way.

**Weaknesses:**

- While the focus on the downstream application of inference-time alignment is nice and leads to powerful results, I'm expecting work that suggests a new way to train a reward model to first evaluate how well the reward model itself is. Do the proposed reward models achieve better prediction capabilities than trajectory-level reward models? same level? I'm suggesting using rewardbench [1] for such evaluation, but even a classic train-test split on some preference dataset will be interesting.

- The author mentioned some prior work that distilled the reward function to a value function to be used in inference-time alignment [2]. Although these methods require an additional training step it is not a real downside, and I believe it should added as a baseline.


[1] Lambert, Nathan, et al. "Rewardbench: Evaluating reward models for language modeling." arXiv preprint arXiv:2403.13787 (2024).
[2] Han, Seungwook, et al. "Value Augmented Sampling for Language Model Alignment and Personalization." arXiv preprint arXiv:2405.06639 (2024).

**Questions:**

- Can training-time alignment algorithms, like PPO, also benefit from the token-level reward? It makes credit assignment easier than trajectory-level rewards.

- [1] is another work that also deals with learning token-level information as part of the reward function. Although it uses different parameterizations, I believe it should be added to the related work section.

- Can you add confidence intervals to the results in Table 1? It will help the reader understand if something like a 48% win rate is significant or not.

[1] Nath, Vaskar, et al. "Learning Goal-Conditioned Representations for Language Reward Models." arXiv preprint arXiv:2407.13887 (2024).

---

> ### Author Response · Authors · 2024-11-21
>
> We thank Reviewer Jvd2 for the detailed feedback and for recognizing our key insight into the limitations of existing methods that use reward models not explicitly trained for token-level decoding. We also appreciate the positive comments on our paper's clarity and theoretical analysis. Below, we address the concerns in detail.
>
> ---
>
> >Weakness 1: While the focus on the downstream application of inference-time alignment is nice and leads to powerful results, I'm expecting work that suggests a new way to train a reward model to first evaluate how well the reward model itself is. Do the proposed reward models achieve better prediction capabilities than trajectory-level reward models? same level? I'm suggesting using rewardbench [1] for such evaluation, but even a classic train-test split on some preference dataset will be interesting.
>
>
> Thank you for the insightful suggestions! We have updated our paper (lines 342 and 398) to include the reward accuracies of both the Autoregressive RM and the traditional trajectory-level RM on the test splits of HH-RLHF and UltraFeedback. Our results show that the proposed Autoregressive RM achieves **the same level** of accuracy to the trajectory-level RM:
>
> * HH-RLHF: Autoregresive RM = 77.6% and trajectory-level RM = 78.3%
> * UltraFeedBack: Autoregresive RM = 75.1% and trajectory-level RM = 75.4%
>
> It is important to note, however, that the best way to assess the quality of an RM for test-time alignment is to evaluate the quality of the generated responses. This has already been thoroughly analyzed in our paper.
>
> ---
>
> >Weakness 2: The author mentioned some prior work that distilled the reward function to a value function to be used in inference-time alignment [2]. Although these methods require an additional training step it is not a real downside, and I believe it should added as a baseline.
>
>
> > Although it requires an additional training step it is not a real downside
>
> One potential downside of distilling a trajectory-level RM into a value function, as done in [2], is the risk of **error accumulation**. The trajectory-level RM itself introduces some error (e.g, accuracy = 75%), and the additional step of distilling it into a value function can compound these errors. In contrast, GenARM avoids this issue by directly learning token-level reward signals from the data, eliminating the need for a separate distillation step and reducing the potential for cumulative errors.
>
> >I believe it should added as a baseline.
>
> We attempted to run the code provided by [2] but were unable to do so successfully. After contacting the authors, they informed us that VAS is in the process of being integrated into the official TRL codebase and suggested we try an example that is still under development. Despite our efforts, we encountered issues with the code. We are currently awaiting the authors' response and appreciate their assistance. Once the complete code becomes available, we will include VAS as a baseline in our work.
>
> [2] Han, Seungwook, et al. "Value Augmented Sampling for Language Model Alignment and Personalization." 2024.
>
> ---
>
> >Question 1: Can training-time alignment algorithms, like PPO, also benefit from the token-level reward? It makes credit assignment easier than trajectory-level rewards.
>
> Yes, reinforcement learning algorithms like PPO can benefit from dense, token-level rewards, as highlighted in prior work [1,2]. We have updated our related work section (lines 127–134) to include a discussion on these approaches and their use of dense rewards for **training** LLMs with RL.
>
> [1] Ng, Andrew Y., Daishi Harada, and Stuart Russell. "Policy invariance under reward transformations: Theory and application to reward shaping." Icml. Vol. 99. 1999.
> [2] Yang, Shentao, et al. "Preference-grounded token-level guidance for language model fine-tuning." Neurips 2023.
>
> ---
>
> >Question 2: [1] is another work that also deals with learning token-level information as part of the reward function. Although it uses different parameterizations, I believe it should be added to the related work section.
>
> Thank you for highlighting this interesting paper on token-level information from a representation perspective! We have updated our related work section to include it.
>
> [1] Nath, Vaskar, et al. "Learning Goal-Conditioned Representations for Language Reward Models." arXiv preprint arXiv:2407.13887 (2024).
>
> ---
>
> >Question 3: Can you add confidence intervals to the results in Table 1? It will help the reader understand if something like a 48% win rate is significant or not.
>
> Thank you for the suggestions! We have updated Table 1 to include confidence intervals (based on three experiments), and the conclusions remain consistent with our original results.
>
> ---
> Thank you again for your time and effort in reviewing our paper! Please let us know if the above explanations do not address your concerns. We are happy to answer any further questions.

---

### Official Review · Reviewer_9RAr · 2024-11-04

**Soundness:** 4
**Presentation:** 4
**Contribution:** 3
**Rating:** 6
**Confidence:** 3

**Summary:**

This paper introduces GenARM, a testing-time alignment method that leverages the Autoregressive reward model, which is designed by the authors to predict next-token rewards for efficient and effective autoregressive generation. The authors show that this new design can allow us to efficiently guide frozen LLMs toward the target optimal distribution that is usually learned by training time alignment methods such as DPO or PPO. Experiments show that GenARM can outperform previous test-time alignment methods significantly without high costs of training larger models. Further, GenARM also supports multi-objective alignment, which allows users to balance trade-offs of different objectives without retraining.

**Strengths:**

- The GenARM's design is relatively simple and straightforward, which allows the model to learn the reward effectively.
- Compared with previous testing time alignment methods that naively used trajectory-based RM to guide the frozen LLMs to generate samples of target distributions, GenARM can provide relatively more accurate token-level guidance and signals with only a small costs to train the GenARM model;
- Empirical experiments shows that GenARM achieves much better performance without training the larger model, while the performance is close to high cost training time-based methods such as DPO;
- The simplicity of the GenARM method also allows users to align the frozen LLMs with multiple objectives, which provide flexibility for real-world alignment problems.

**Weaknesses:**

- Compared with training time-based method, there is still gap of the current methods and DPO, while theoretically the authors show that the method can approximate the target optimal distribution;
- The evaluation experiments are relatively simple, only conducted on simple benchmarks;
- It would be good to compare [1] since this is also closely related to testing-time alignment.
- It would be good to add some discussion of previous token-level reward methods or literature, e.g. [2]




[1] Cascade Reward Sampling for Efficient Decoding-Time Alignment. https://arxiv.org/pdf/2406.16306
[2] Preference-grounded Token-level Guidance for Language Model Fine-tuning

**Questions:**

- There are two ways to predict or output token level rewards, one is to directly view the problem as token-level prediction, and another method is to use an additional head to predict the real-values of the reward, I wonder if authors tried and compare these two methods?

- Regarding the gap of training time methods, I wonder if we leverage more advanced sampling methods such as MCTS to generate more trajectories, will the gap be eliminated?

---

> ### Author Response · Authors · 2024-11-21
>
> We thank Reviewer 9RAr for the detailed and thoughtful feedback. We are pleased that the reviewer finds our method's design simple and effective, our test-time approach more accurate and cost-efficient than existing test-time alignment baselines, and our weak-to-strong guidance results promising. We also appreciate reviewer's recognition of our method's simplicity in handling multiple objectives. Below, we address the reviewer's concerns in detail.
>
> ---
>
> >Weakness 1: Compared with training time-based method, there is still gap of the current methods and DPO, while theoretically the authors show that the method can approximate the target optimal distribution;
>
> To clarify, when the reward model (RM) and the base LLM are of the same size, our proposed GenARM **matches the performance** of the training-time method DPO (there is no performance gap). In contrast, other test-time alignment baselines, such as ARGS and TransferQ, perform worse than DPO in this setting. Notably, GenARM effectively closes this performance gap.
>
> When the RM is smaller than the base LLM (e.g., RM is 7B while the base LLM is 70B), GenARM does exhibit a performance drop compared to DPO. However, it is important to emphasize that GenARM requires training only the smaller RM (e.g., 7B), whereas training the much larger LLM (e.g., 70B) in DPO may be **computationally prohibitive or entirely infeasible**. This makes GenARM a practical and scalable alternative in such scenarios.
>
>
> As for the theoretical results: We show that the proposed Autoregressive RM is expressive enough as a function class to represent any decoding distribution. This **expressiveness is crucial**; using a parametrization that cannot capture the target distribution would undermine the entire test-time alignment approach. However, this result **does not** guarantee how well such a function can be learned from the data. Empirically, larger LLMs with more pretrained knowledge provide a better initialization for reward models, leading to more accurate RMs after fine-tuning. Due to computational constraints, we only trained 7B RMs in this work, but we anticipate that training a larger RM could further enhance performance.
>
> ---
>
> >Weakness 2: The evaluation experiments are relatively simple, only conducted on simple benchmarks;
>
> Our evaluation experiments are sufficiently comprehensive to demonstrate the effectiveness of GenARM in various scenarios, including comparisons with test-time and training-time baselines, standard reward-guided decoding, and practical applications like weak-to-strong guidance and multi-objective alignment. Specifically:
>
> 1. Standard Reward-Guided Generation on general preference (Section 6.1):
>     * Conducted on the widely used HH-RLHF dataset with RM and LLM of the same size.
>     * Results show that:
>         * GenARM outperforms other test-time alignment baselines.
>         * GenARM is more efficient during inference.
>         * It achieves comparable performance to training-time alignment methods.
> 2. Weak-to-Strong Guidance (Section 6.2):
>     * Evaluated on the widely used AlpacaEval2 benchmark.
>     * Results show that:
>         * GenARM outperforms test-time alignment baselines.
>         * It performs comparably to training-time methods when RM and LLM are of the same size.
>         * GenARM effectively provides weak-to-strong guidance, recovering most of the performance gap with training-time methods while requiring only the smaller RM. This is crucial in scenarios where training-time methods are computationally infeasible.
> 3. Multi-Objective Alignment (Section 6.3):
>     * Conducted on the widely used multi-objective dataset PKU-SafeRLHF.
>     * Results highlight that:
>         * GenARM is efficient and effective in aligning with multi-dimensional preferences.
>         *  GenARM enables weak-to-strong guidance in multi-objective alignment.
>
> These experiments, leveraging widely recognized benchmarks, effectively showcase the versatility and practicality of GenARM across real-world alignment challenges.

---

> > ### Author Response · Authors · 2024-11-21
> >
> > >Weakness 3: It would be good to compare [1] since this is also closely related to testing-time alignment.
> >
> > We have revised our paper to include CARDS [1] in the related work section (Section 2, line 114-137) and compare it with our method in the experiments (Section 6.1: Table 1 and Table 2). Unlike our approach, [1] **does not use a token-level RM** to guide text generation. Instead, it repeatedly generates small semantic segments (a few tokens) and evaluates them using a **trajectory-level RM** to select the most satisfactory option. Similar to ARGS [3] (which we already cited and compared with in the original version), [1] relies on trajectory-level RMs to evaluate partial responses, which can be inaccurate. Moreover, it induces significant inference costs as it needs to generate several candidate segments before choosing one. In contrast, our method is more efficient during inference, as the Autoregressive RM directly predicts the next-token reward for next-token generation.
> >
> > Experimentally, GPT-4 evaluation on the HH-RLHF dataset demonstrates that our proposed GenARM achieves a **higher win rate** compared to CARDS [3], as summarized below:
> >
> > | Method | Win | Tie | Lose | Win + $\frac{1}{2}$Tie |
> > |-|-|-|-|-|
> > |GenARM VS CARDS|54.67|5.22|40.11|57.27|
> >
> > Moreover, GenARM is significantly **more efficient** than CARDS during inference, as evidenced by the inference time required to generate 128 tokens using a 7B base LLM and a 7B RM, as detailed below.
> >
> > |  | ARGS | GenARM | Transfer-Q | CARDS |
> > |-|-|-|-|-|
> > |Inference time (s) |7.74|7.28|130.53|87.09|
> >
> > Therefore, our conclusion remains unchanged: our method achieves the highest win rate and the lowest inference cost among all test-time alignment approaches.
> >
> >
> > >Weakness 4: It would be good to add some discussion of previous token-level reward methods or literature, e.g. [2]
> >
> > Thank you for highlighting [2] on dense rewards! We have updated our related work section (Section 2, line 114-137) to incorporate and discuss these contributions, as detailed below:
> >
> > We acknowledge that [2], highlighted by the reviewers, are pivotal works leveraging dense, token-level reward signals for **training** LLM policies with reinforcement learning (RL). These approaches tackle the well-known challenges of learning from sparse and delayed reward signals in RL [4]. However, our work differs in several key aspects:
> >
> > 1. Our paper focuses on **test-time** alignment, specifically without requiring any training of the LLM policy network. Unlike [2], which use dense reward signals for RL training, we leverage a token-level RM to enable efficient next-token prediction directly at test time.
> > 2. Theoretically, we have proven that the parametrization of our proposed Autoregressive RM can guide frozen LLMs toward any distribution achievable by traditional RMs within the KL-regularized RL framework. This result is **fundamental for enabling test-time alignment** and is a capability **not addressed** in prior works that focus on providing dense rewards for training.
> >
> >
> > [1] Li, Bolian, et al. "Cascade reward sampling for efficient decoding-time alignment." 2024.
> > [2] Yang, Shentao, et al. "Preference-grounded token-level guidance for language model fine-tuning." Advances in Neural Information Processing Systems 36 (2023).
> > [3] Khanov, Maxim, Jirayu Burapacheep, and Yixuan Li. "ARGS: Alignment as Reward-Guided Search." ICLR 2024.
> > [4.] Ng, Andrew Y., Daishi Harada, and Stuart Russell. "Policy invariance under reward transformations: Theory and application to reward shaping." Icml. Vol. 99. 1999.

---

> > > ### Author Response · Authors · 2024-11-21
> > >
> > > >Question 1: There are two ways to predict or output token level rewards, one is to directly view the problem as token-level prediction, and another method is to use an additional head to predict the real-values of the reward, I wonder if authors tried and compare these two methods?
> > >
> > > In this work, we frame the problem as token-level prediction to allow the model to predict next-token rewards, which is convenient for next-token generation. This approach **already employs a linear head to compute reward values**, as dictated by the LLM architecture. For example, given a partial response "I like," the LLM uses its representation of "I like" and applies a linear head (and softmax) to map it to the vocabulary probabilities.
> > >
> > > Therefore, when computing next-token rewards (which requires predicting rewards for all tokens in the vocabulary), the two methods mentioned by the reviewer are effectively equivalent, as both rely on mapping representations to reward values using a linear transformation.
> > >
> > > >Question 2: Regarding the gap of training time methods, I wonder if we leverage more advanced sampling methods such as MCTS to generate more trajectories, will the gap be eliminated?
> > >
> > > Leveraging advanced search methods like MCTS to enhance performance with our Autoregressive RM is an interesting future direction that could potentially improve text generation quality further!
> > >
> > > However, it is important to highlight that advanced search algorithms, such as MCTS, typically come with significant inference costs. **A key advantage of GenARM is its ability to perform well without relying on expensive search methods**. With simple next-token decoding, GenARM matches the performance of training-time baselines when the RM and LLM are of the same size.
> > >
> > > In fact, all of our test-time alignment baselines including ARGS, CARDS, Transfer-Q are essentially advanced sampling/search methods, using trajectory-level RM to search for responses with high reward. GenARM not only outperforms these baselines but does so efficiently, avoiding the computational overhead of complex search algorithms. This efficiency makes GenARM a practical and scalable solution for test-time alignment.
> > >
> > > ---
> > > Thank you again for your time and effort in reviewing our paper! Please let us know if the above explanations do not address your concerns. We are happy to answer any further questions.

---

> > > > ### Author Response · Authors · 2024-11-24
> > > >
> > > > We sincerely thank the reviewer for your time and effort in reviewing our paper! We have addressed the specific points you raised regarding the implications of our theoretical results, the evaluation settings, additional test-time alignment baselines and related work discussion.
> > > >
> > > > Please let us know whether we have fully addressed your concerns. We are more than happy to provide additional clarifications if you have further questions. Thank you!

---

### Author Response · Authors · 2024-11-21
**Global Response**

We thank all reviewers for their valuable feedback. The reviewers recognized our approach as novel (S14E), effective in addressing prior limitations (Jvd2), and paper well-written (Jvd2, S14E). They appreciated our comprehensive experiments (S14E), theoretical analysis (Jvd2), and the simplicity and effectiveness of our method (9RAr). Additionally, they commended its versatility across applications like standard text generation, weak-to-strong guidance, and multi-objective alignment (Dm7h, 9RAr), and found our work timely, promising (Dm7h, Jvd2), and cost-efficient compared to test-time baselines (9RAr).

Below, we address common concerns and misunderstandings and provide additional experimental results. We also outline the updates made to the paper.

---

### Concern 1: Related work on dense rewards

Reviewers Jvd2, 9RAr, and Dm7h highlighted several prior works on dense rewards in training LLMs that were missing from our related work section. We have updated Section 2 (lines 114–137) to incorporate these papers. Below, we summarize the most relevant related works:

We acknowledge that [1,2], highlighted by the reviewers, are pivotal works leveraging dense, token-level reward signals for **training** LLM policies with reinforcement learning (RL). However, our work differs in several key aspects:

1. Our paper focuses on **test-time** alignment, specifically without requiring any training of the LLM policy network. Unlike [1,2], which use dense reward signals for RL training, we leverage a token-level RM to enable efficient next-token prediction directly at test time.
2. Theoretically, we have proven that the parametrization of our proposed Autoregressive RM can guide frozen LLMs toward any distribution achievable by traditional RMs within the KL-regularized RL framework. This result is **fundamental for enabling test-time alignment** and is a capability **not addressed** in prior works that focus on providing dense rewards for training.

References:
[1] Feng, Yihao, et al. "Fantastic Rewards and How to Tame Them: A Case Study on Reward Learning for Task-oriented Dialogue Systems." ICLR 2023
[2] Yang, Shentao, et al. "Preference-grounded token-level guidance for language model fine-tuning." Neurips 2023.

---

### Concern 2: Comparison with another test-time alignment method

Reviewers 9RAr and Dm7h highlighted another test-time alignment method, CARDS [3], and Reviewer Dm7h expressed concern that it might use a similar token-level reward approach. We have revised our paper to include [3] in the related work section (Section 2, line 114-137) and compare it in the experiments (Section 6.1: Table 1 and Table 2).

To clarify, unlike our method, **CARDS does not use token-level RM and suffers from significant inference costs**. Instead, it repeatedly generates small semantic segments and evaluates them using a **trajectory-level RM** to select the most satisfactory option. Similar to ARGS [4] (which we already compared with), [3] relies on trajectory-level RMs to evaluate partial responses, which can be inaccurate. Moreover, it induces significant inference costs as it needs to generate several candidate segments before choosing one. In contrast, our method is more efficient during inference, as the Autoregressive RM directly predicts the next-token reward for next-token generation.

Experimentally, GPT-4 evaluation on the HH-RLHF dataset demonstrates that our proposed GenARM achieves a **higher win rate** compared to CARDS, as summarized below:

| Method | Win | Tie | Lose | Win + $\frac{1}{2}$Tie |
|-|-|-|-|-|
|GenARM VS CARDS|54.67|5.22|40.11|57.27|

Moreover, GenARM is significantly **more efficient** than CARDS during inference, as evidenced by the inference time required to generate 128 tokens using a 7B base LLM and a 7B RM, as detailed below.

|  | GenARM | CARDS |
|-|-|-|
|Inference time (s) |7.28|87.09|

References:
[3] Li, Bolian, et al. "Cascade reward sampling for efficient decoding-time alignment." arXiv preprint arXiv:2406.16306 (2024).
[4] Khanov, Maxim, Jirayu Burapacheep, and Yixuan Li. "ARGS: Alignment as Reward-Guided Search." ICLR (2024).

----
### Paper updates

All the updates in the paper are in blue.

* (JVd2, 9RAr, DM7h) Additional discussion on prior literature on dense rewards in Section 2 (lines 114–137).
* (9RAr, DM7h) Adding experimental comparison with CARDS in Section 6.1 (Table 1, Table 2.)
* (JVd2) Adding confidence intervals to Table 1.
* (JVd2) Adding results on reward model accuracies in Section 6.1 (lines 342) and Section 6.2 (lines 398).
* (S14E) Adding results on comparison with BoN in Section 6.1 (Table 1)
* (DM7h) Moving parts of related work sections to Appendix A (lines 717-738).
* (S14E) Adding training-time comparison in Appendix C.1 (lines 803-817).


---
We thank all reviewers again for the time and effort in reviewing our paper! Please let us know if the above explanations do not address your concerns. We are happy to answer any further questions.

---

### Meta-Review · Area_Chair_VR9e · 2024-12-21

**Metareview:**

This paper proposes autoregressive reward models for test-time alignment. Given preference data, it trains a reward model that can predict a reward for each token as $log p_r(y_t \mid x_t, y_{<t})$, and the total reward is given by the sum of these individual rewards. It then trains the reward model using the standard approach of maximizing the log-likelihood assuming the Bradley-Terry distribution. At test time, we generate tokens using a product of the base model's probability multiplied by an adjusted reward model $p_r$ score.

However, once the reward model is trained, the base LLM isn't trained but instead is guided at test time using the learned token-level reward model. So, why is this better than training the base LLM directly using DPO? Authors claim that when the reward model is a weaker LLM, this is more computationally efficient for training.

Strengths:
1. Novel approach for test-time alignment. The most useful fact is that many people are not releasing trained reward models. This can allow someone to quickly try these reward models with any LLM to get quick numbers without having to train the reward model -- something that not everyone can do. This is a highly useful fact!

2. Authors were able to add experiments with the CARDS baseline and show that their approach does better.

3. Reviewers found the approach simple and easy to use. This matters for real-world usage.

Weakness:
1. When using a weaker reward model, the performance does not match that of using a DPO trained policy. However, the authors do show that when using the same reward model class as the LLM, the performance does match. But when you do this, you might as well train the LLM directly using DPO as the computational benefit of avoiding training a big model is lost. Further, there is some overhead at inference due to computing rewards as well. For this reason, I think labs that can afford to train a model and have fixed preference data will not find this approach useful.

2. The reviewer suggested experiments with the value distillation approach that the authors were unable to add. This would have been nice.

Overall, a good approach. There are some trade-offs here that I mentioned above. The real audience of this paper will be people with fewer resources who can only train small reward models, or who can benefit from large GenARM reward models being made available and who can quickly try these reward models with **any** LLM, that has same vocabulary, without having to do expensive RLHF. This is an important advantage and which is why I recommend accepting.

**Additional Comments On Reviewer Discussion:**

Reviewers raised the following main concerns:

1. Missing related work (raised by Jvd2, 9RAr, and Dm7h): The authors were able to add the related work and resolve this.

2. Comparison with another test-time alignment approach: Reviewers 9RAr and Dm7h suggested comparing with the CARDs approach. Authors provided this number showing gains along with noting that the CARDs approach is computationally expensive as it requires generating sub-trajectories and using a trajectory-level reward. Reviewer Jvd2 suggested adding the VAS baseline that distills the trajectory-level reward into a value, but the authors were unable to run this. That said, they noted that this reward distillation step can lead to error accumulation as the trajectory-level reward already has an error.

3. Noting the gap in performance between the proposed GenARM test-time alignment approach vs using training the policy with DPO: Authors noted that the gap disappears when the reward model is the same class as the model but this isn't very convincing as then you also loose the computational benefit of GenARM. However, for people with limited resources, training the full model may not be possible.

I think the authors addressed the concerns reasonably well. I would encourage authors to add the VAS baseline and show numbers with VAS and CARDs across all domains.

---

### Decision · Program_Chairs · 2025-01-22

Accept (Poster)